# Effective Coping with Academic Stress Is a Matter of Personality Types: Revisiting the Person-Centred Approach

**DOI:** 10.3390/bs13080687

**Published:** 2023-08-18

**Authors:** Cristina Varo, María del Mar Aires-González, María García-Jiménez, María Eva Trigo, Francisco Javier Cano-García

**Affiliations:** 1Department of Personality, Assessment, and Psychological Treatments, Universidad de Sevilla, 41018 Seville, Spain; cvaro@us.es (C.V.); maires@us.es (M.d.M.A.-G.); 2Department of Experimental Psychology, Universidad de Sevilla, 41018 Seville, Spain; mgarciaj@us.es (M.G.-J.); trigo@us.es (M.E.T.)

**Keywords:** personality types, coping strategies, perceived coping efficacy, academic stress, gender

## Abstract

Recent literature provides alarming data on the increase in university academic stress. The role of personality in understanding and addressing this problem is well established. However, this evidence could be improved by adopting a person-centred approach (e.g., types), as opposed to the usual variable-centred approach (e.g., traits), and considering the role of gender. Our aim was to explore how personality types and gender relate to coping strategies and perceived coping efficacy for academic stress. A total of 810 university psychology students completed the NEO-FFI Inventory and the Coping Strategies Inventory. Post hoc tests for MANOVA and ANOVA were performed. Types and gender were used as predictors and coping strategies, and perceived coping efficacy as criteria. There was no type-gender interaction. Types combining low neuroticism-high conscientiousness (e.g., entrepreneur) chose the most adaptive coping strategies and showed the highest levels of perceived coping efficacy, while high neuroticism-low conscientiousness types (e.g., insecure) opted for maladaptive coping strategies and presented the lowest perceived coping efficacy. Gender was not associated with perceived coping efficacy but with use (e.g., women prefer emotional expression). The personality typology provided useful information on individual differences in coping with academic stress, which can help guide specific strategies to manage it.

## 1. Introduction

Personality can be defined as “an individual’s unique variation on the general evolutionary design for human nature, expressed as a developing pattern of dispositional traits, characteristic adaptations, and integrative life stories complexly and differentially situated in culture” (p. 212) [1]. Traits, as basic behavioural tendencies, are the core of personality, and nowadays, the Five Factors Models (FFM) are the most widely accepted and commonly used trait theories of personality [2]. FFM defines personality based on five traits: extraversion, degree of sociability, positive emotionality and general activity (e.g., I enjoy talking to people); agreeableness, altruistic, sympathetic and cooperative tendencies (e.g., I tend to think the best of people); conscientiousness, one’s level of self-control in planning and organisation (e.g., I work hard to achieve my goals); neuroticism, the tendency to experience negative emotions and psychological distress in response to stressors (e.g., when I am under heavy stress, I sometimes feel like I am going to fall apart); and openness to experience, levels of curiosity, independent judgment and conservativeness (e.g., I have a wide variety of intellectual interests).

The issue with studies on personality and stress coping strategies resides in the isolated approach to traits [3]. In other words, the effect of high and low scores for single traits (e.g., extraversion or neuroticism) is prioritised, creating a variable-centred approach rather than assessing high-low combinations of different traits simultaneously (e.g., high extraversion, low neuroticism) in a person-centred approach [4]. Personality psychology has traditionally focused on the variable-centred approach, which looks at individual differences and their observable variations in different personality traits. However, this may overlook one essential aspect of personality: the way in which traits are configured within an individual. For example, an individual is not exclusively extraverted, conscientious, or neurotic, but rather a combination of these traits. The score of any single trait may strengthen or weaken the relationship between another one [3,4].

Thus, one of the motivating assumptions of the person-centred approach is the idea that traits should not be studied in isolation. The person-centred approach identifies groups of individuals who share particular or similarly connected attributes [5]. In the field of personality, the person-centred approach identifies individuals with the same basic personality profile. Personality types describe personality by assessing an individual’s scores on several personality dimensions.

In accordance with Donnellan et al. [6], a replicable and empirically validated personality typology can play an important and even necessary role in research on personality development. Nowadays, there is little evidence for discrete personality types. However, in recent times, there has been a resurgence of interest in this approach. Fisher and Robie [4] have derived an empirical typology on the responses to an FFM measure of more than 3 million people worldwide. From scores on four of the big five, all except openness to experience, they derived three latent profiles labelled maladaptive, adaptive, and highly adaptive. A systematic review of FFM with a person-centred approach has noted the lack of a definitive solution [3]. The included studies had obtained between two and five latent profiles, with the majority finding three or four. Neuroticism and openness to experience were, respectively, the most and least useful traits in obtaining profiles. Irrespective of the number of profiles obtained, their association with perceived well-being has been found [7]. Person-centred approaches based on other personality models have also obtained relevant results in academic contexts. For example, one based on Cloninger’s biopsychosocial model found profiles associated with different degrees of academic engagement in pre-university students [8]. Another study found five personality profiles based on the HEXACO model of personality in American college students [9]. Different profiles have been associated with different levels of perceived well-being, with more stability over time of distress than well-being, which emphasises the importance of intervention with these students [10].

There is an older, theoretically derived typology of traits, which serves as inspiration because it addresses coping with the academic stress of university students, as will be detailed below. Torgersen [11] explored eight personality types based on different high vs. low combinations of the hysterical (extraversion), oral (neuroticism) and obsessive (conscientiousness). Vollrath and Torgersen [12] replicated these eight personality types (see Table 1 for the composition of the types and their labels), which may serve as useful and convenient labels for trait combinations associated with consequential outcomes.

Lazarus and Folkman [13] described coping as cognitive and behavioural reactions employed to deal with stressful demands perceived as exceeding one’s personal resources. Traditionally, there have been two views of coping. On the one hand, coping styles conceive coping as a stable and consistent individual difference; on the other, coping strategies emphasise the coping process as a dynamic person-context transaction [14]. In the terms in which we described personality at the beginning of the introduction, the strategy view understands coping as a characteristic adaptation, while the style view understands coping as a trait [1]. Tobin et al. [15] determined that coping could be organised into two general categories; “coping activities that engage the individual with, and coping activities that disengage the individual from, the stressful situation” (p. 355). The authors differentiated between four specific engaged coping strategies: problem-solving, understood as cognitive and behavioural strategies aimed at eliminating stress by modifying the situation which produces it (e.g., “I tried hard to resolve the problem”); cognitive restructuring, understood as cognitive strategies that modify the meaning of the stressful situation (e.g., “I convinced myself that things were not as bad as they seemed”); social support: understood as strategies referring to the search for emotional support (e.g., “I found somebody who was a good listener”); and emotional expression, defined as strategies aimed at releasing the emotions that arise during the stressful process (e.g., “I analyse my feelings and just let them out”). They noted that four specific disengaged coping strategies could also be employed: problem avoidance, understood as strategies that include denial and avoidance of thoughts or acts related to the stressful event (e.g., “I refused to think about it too much”); wishful thinking, understood as cognitive strategies that reflect the wish that reality was not stressful (e.g., “I wished I could have changed what had happened”); social withdrawal, defined as strategies to withdraw from friends, family, peers and significant others associated with the emotional reaction to the stressful process (e.g., “I spent some time by myself”); and self-criticism, understood as strategies based on self-blame and self-criticism due to the occurrence or inadequate handling of the stressful situation (e.g., “It was my mistake, so I have to suffer the consequences”). Likewise, engaged coping strategies have been treated as adaptive coping, while avoidance (disengaged) coping strategies are viewed as forms of maladaptive coping [16,17,18]. It is adaptive to adjust the choice of coping strategy according to the present controllability of the stressor, so the use of problem-focused coping strategies for controllable stressors is associated with lower levels of stress [19]. In the university context, it has been shown that academic stress can lead to academic burnout, and this relationship is mediated by coping [20], that coping predicts up to 33% of the variance in life satisfaction [21], that maladaptive coping mediates the association between stress and suicidal ideation [22], and that avoidant coping is associated with emotional eating [23].

The FFM has contributed significantly to understanding coping. Most research has focused on the role personality plays in coping, particularly traits like neuroticism and extraversion, while other personality dimensions have received relatively less attention [24]. Studies have found that individuals with high neuroticism experience more stressful events and use passive and maladaptive coping, such as ignoring the problem, distracting, venting, or avoiding. Meanwhile, individuals with higher extraversion use more active coping strategies and seek more social support. However, regarding other personality dimensions, openness to experience and conscientiousness are related to active and less evasive coping [25]. Milad and Bogg [26] found that the Big Five, especially conscientiousness and extraversion, were associated, through coping, among other processes, with physiologically measured allostatic load 10 years later. In the university context, academic resilience has been positively associated with problem-focused coping, extraversion, openness to experience, agreeableness, and conscientiousness, and negatively associated with emotion-focused coping, avoidant coping, and neuroticism [27]. Evans et al. [28] showed direct and indirect associations across stress coping of three of the Big Five with subjective happiness and academic satisfaction in pre-university students. Specifically, extraversion and neuroticism directly predicted subjective happiness, respectively, in a positive and negative sense, while conscientiousness was directly and positively associated with academic satisfaction. Conscientiousness and extraversion were also positively associated with academic satisfaction using productive coping, while neuroticism was associated with unproductive coping without this association mediating the association with subjective happiness. Agreeableness and openness did not contribute to the prediction.

In the extensive literature on academic stress in university students [29,30,31], alarming trends have recently been detected in perceived stress in this community. University students are exposed to a wide range of potentially stressful situations that can negatively affect their academic achievement and health [32]. In fact, high-stress levels experienced by university students are considered one of the more prevalent psychosocial problems in the university community. In their daily lives, university students must manage a wide variety of demands, both academic and non-academic [33]. Therefore, the university is a stressful time for most students. According to Russell and Petrie [34], a student’s ability to adapt to university stress depends on three factors: academic performance, social adjustment, and personal adjustment. Yet, little is known about how personality types relate to stress and coping, particularly among a student university sample. Only one study has examined how personality types compare in terms of the experience of stress and emotions and coping strategies in a sample of university students [12]. The authors found that the most favourable stress and coping profile corresponded to personality types with low neuroticism and high conscientiousness (entrepreneur and sceptic), whereas those in which high neuroticism combined with low conscientiousness (impulsive and insecure) showed high vulnerability to stress and poor coping. Furthermore, the authors noted that the effects of extraversion were more ambiguous and appeared to depend on the specific combinations of neuroticism and conscientiousness. In general, high levels of extraversion strengthened the adaptive coping of low neuroticism and high conscientiousness types but did not enhance or even worsen the opposite [12]. However, after several decades of studies on the impact of stress and coping strategies, substantial gaps and inconsistencies remain [16].

Secondly, the literature reiterates the importance of considering both coping strategies and perceived coping efficacy in protection against stress [32]. Though stress coping skills and perceived coping efficacy are two different concepts, they are related. Perceived coping efficacy could be considered a specific component of self-efficacy. Self-efficacy has been described as the belief in one’s capability to produce designated levels of performance for events that affect one’s life, “an important factor in determining how people feel, think, motivate themselves and behave” [35]. Self-efficacy seems to strongly influence university adjustment [35,36] since individuals with a high sense of self-efficacy tend to feel able to overcome situational difficulties, attributing little stress to them. Different studies have found an association between the perception of self-efficacy and the coping strategies university students employ [32,37]. Coping strategies such as problem-solving, positive re-evaluation and the search for social support have a positive effect on self-efficacy; however, other coping strategies, such as venting negative emotions and negative auto-focus, have a detrimental effect [37].

Finally, gender is an essential variable in the relationship between university students and stress. Previous research has noted gender differences in stress levels and coping strategies [38,39,40,41,42]. Though women report higher stress levels than men [38], they also make better use of emotional support [41,42]. Men, in contrast, tend more toward avoidance-focused coping [39] and problem-focused coping than women [41]. The only reference we have on coping with academic stress as a function of personality types is Vollrath and Torgersen’s [12]. On the one hand, they found gender differences in the prevalence of personality types, with more males in the spectator, sceptic and hedonist types and more females in the brooder, impulsive and complicated types. However, they found no interactive gender-type effects in relation to coping.

To date, except for Vollrath and Torgersen [12], no study has investigated the association between personality types, coping strategies for academic stress, and perceived coping efficacy among university students. A better understanding of how these variables relate would have implications in the fields of education and clinical psychology. The use of adaptative coping strategies combined with a high level of perceived coping efficacy could result in greater well-being, quality of life, adaptation, or adjustment at university. Considering these gaps in the literature, the main aim of the present study was to explore the associations of eight personality types based on Torgersen’s typology with academic stress, coping strategies, and perceived coping efficacy in a large sample of university students. Second, we explored the gender distribution of personality types and coping strategies.

## 2. Materials and Methods

### 2.1. Participants

Data was collected from 1305 psychology students at the University of Seville (Spain) who voluntarily participated. After the personality types analysis, which required the elimination of the mean levels in the scores, as described in the Procedure section, the final sample included 810 participants, including 643 women (79.4%) and 167 men (20%); the average age was slightly lower for women (*M* = 20.09; *SD* = 2.991) than for men (*M* = 20.81; *SD* = 5.261), *p* = 0.021.

### 2.2. Measures

Personality was assessed using the Spanish version of the Five-Factor Inventory (NEO-FFI) [43,44]. The NEO-FFI is a self-report inventory that can be completed in approximately 15 min and measures the five personality dimensions described by Costa and McCrae: extraversion, agreeableness, conscientiousness, neuroticism, and openness to experience. It is composed of 60 items (12 per domain) rated on a 5-point Likert scale by indicating to what extent the respondents agree with each of the statements regarding themselves (0 = strongly disagree, 1 = disagree, 2 = neither agree nor disagree, 3 = agree, 4 = strongly agree). Scores for each domain are the sum of the responses to the 12 items. The NEO-FFI has shown adequate levels of validity and reliability across a range of diverse populations. This Spanish version has appropriate indices of reliability and validity. Each of the five domains has been found to possess adequate internal consistency (α = 0.71 to 0.82) [44] (see Appendix A in the Appendix A for more details about the reliability of the scale).

Coping was assessed with the Coping Strategies Inventory (CSI) originally created by Tobin et al. [15] and adapted to Spanish by Cano-García et al. [45]. This instrument contains a blank space for the respondent to describe a stressful situation in the maximum amount of detail. This is followed by the 40 items of the instrument, which reflect the thoughts, attitudes, feelings and coping behaviours linked to the situation described and scored on a Likert-type 5-point scale, where 0 = strongly disagree and 4 = strongly agree. The instrument is composed of eight subscales, each with five items, so the range of direct scores for each is from 0 to 20. High scores indicate a major use of these strategies when faced with different stressful situations. The subscales are the following: problem-solving (PS), cognitive restructuring (CR), social support (SS), emotional expression (EE), problem avoidance (PA), wishful thinking (WT), social withdrawal (SW), and self-criticism (SC).

Finally, in order to explore the perceived coping efficacy, one additional item is included (“I believe I can cope with the situation”). The variance and Cronbach’s alpha coefficients for the eight primary coping strategies ranged from 0.63 to 0.89 [45], revealing good psychometric properties with Spanish samples. In the current study, the reliability indices for each coping strategy showed acceptable to excellent internal consistency in our sample, ranging from α = 0.71 for the problem avoidance strategy to α = 0.91 for the strategy of self-criticism (see Appendix A in the Appendix A for more details about the reliability of the scale).

### 2.3. Procedure

In the first semester of each academic year from 2010–2011 to 2021–2022, students in the Personality Psychology and Human Diversity, in the second year of the Psychology degree program at the University of Seville (Spain), completed a battery of instruments, including the CSI and the NEO-FFI. The work was performed on campus anonymously under the supervision of teaching staff. The stressful situation to be addressed in the CSI was academic; specifically, students had to describe a stressful academic situation they experienced during their university studies. They then respond to the 40 items of the inventory, plus the item related to perceived coping efficacy.

In order to create the personality type variable, scores for neuroticism, extraversion and conscientiousness were changed to an ordinal variable and assigned a value of high, medium or low. Subjects with medium scores were excluded, and the rest were assessed according to Torgersen’s classification of eight personality types (1995) (See Table 1). However, instead of using the medians to determine the score levels, as those authors did, we transformed the NEO-FFI dimensions using the T-scores from a scale that relied on a large normative Spanish sample [44] in which T-scores means = 50 were considered for each dimension. We classified the participants of our study in low, medium, or high scores following a statistical scoring criterion with ±0.5 standard deviations from the normative group mean.

The research was carried out in accordance with the Declaration of Helsinki, and informed consent was obtained in writing from each participant. Data processing has complied with the current regulations in this regard: first, with the Spanish Personal Data Protection Act of 1999 and, second, with the Spanish Data Protection Rule (GDPR) of 2016, which guarantees the anonymity and security of the information at all times As the subjects were university students, the following requirements were established: all were recruited by faculty members with whom they have had no academic involvement; the activity was voluntary, without incentives of any kind; and the activity was performed outside of class.

### 2.4. Data Analyses

First, the differences in the eight coping dimensions were contrasted in a two-way MANOVA with gender and personality types as independent variables. Beforehand, the assumption of multivariate normality was evaluated with Mahalanobis distances and Mardia’s test; homogeneity of the variance-covariance matrices was assessed using Box’s M test; and the homoscedasticity for each coping strategy was measured with Levene’s tests. Because of the unbalanced design and the lack of normality and homoscedasticity, Wilks’ Lambda was chosen as the contrast statistic [46].

Because no interaction was detected and some of the coping strategies revealed heteroscedasticity, the MANOVA was followed by 16 one-way ANOVA tests, eight with gender as a factor and eight with personality types as a factor. This was to run Welch’s test and Games–Horwell when homoscedasticity was not met, which is not possible in factorial ANOVA. Adjusted probabilities (adjusted *p* = *p* × 16) were used in all the ANOVA tests. The *R*^2^ of each factor was used as the effect size index, considering a medium effect size for values of 0.06 or a large effect size of 0.14. When statistical differences were found, multiple comparison tests, Tukey or Games–Horwell, were run depending on homoscedasticity, with an additional Bonferroni adjustment for 16 tests.

Finally, a one-way ANOVA followed by post hoc comparisons was performed to analyse the relationships between personality types and perceived coping efficacy, using Welch’s *F* and the Games–Horwell test because of the heteroscedasticity. Again, *R*^2^ was used as the effect-size index.

The software SPSS 26.0 was used for the MANOVA and ANOVA plus post hoc multiple comparisons; JASP 0.16 was used for reliability and Mardia’s tests.

## 3. Results

### 3.1. Descriptive Analyses

Table 2 shows the percentage of participants assigned to each personality type. The least frequent category was the spectator type, and the most frequent was the insecure type. There was a significant difference in the distribution of personality types between genders, χ^2^(7, *N* = 810) = 49.94, *p* < 0.001, although the effect size did not reach the medium level, *r*_φ_ = 0.24. The standardised residuals greater than 2.97, *Z* value of *p* = 0.003, were then inspected, applying a Bonferroni adjustment for 16 cells (0.005/16 = 0.003). Only the standardised residuals of spectator men (3.9) and hedonist men (3.8) were higher than expected. Other standardised residuals ranged from 0.3 (impulsive women) to −2.4 (complicated men).

### 3.2. How Personality Types and Coping Strategies Relate

Multivariate normality was analysed by computing Mahalanobis distances and Mardia’s test for skewness and kurtosis. Mahalanobis distances ranged from 1.41 to 29.33 (*M* = 8.99, *SD* = 4.48); only three of the 810 pieces of data were outliers, higher than chi-square (*df* = 8, *α* = 0.001) = 26.13. Mardia’s coefficient (c) of skewness was significant, c = 4.84, χ^2^(120) = 653.21, *p* < 0.001, but the assumption of normality was met for kurtosis, c = 80.67, z = 0.75, *p* = 0.451. The variance-covariance matrices were not homogeneous, Box *M* = 795.99, *F*(504, 21,739.75) = 1.34, *p* < 0.001. The MANOVA did not reveal significant interaction between gender and personality types, Wilks’ Lambda = 0.92, *F*(52, 4243.44) = 1.17, *p* = 0.187, but did reveal large main effects for both genders, Wilks’ Lambda = 0.97, *F*(8, 787) = 3.53, *p* = 0.001, and personality type, Wilks’ Lambda = 0.69, *F*(56, 4243.44) = 3.53, *p* < 0.001.

In the case of emotional expression, univariate homoscedasticity was not met for the analysis of gender; in the case of problem-solving, self-criticism and emotional expression, it was not met for the analysis of personality types. Thus, we substituted Welch’s *F* and Games–Horwell tests for Snedecor’s *F* and Tukey tests. Table 3 shows the results of ANOVA and means for gender, while Figure 1 shows the post hoc comparisons when statistical differences and at least medium effect sizes were found in personality types. As can be observed, gender differences were significant in four dimensions of the CSI; emotional expression, social support, problem avoidance and social withdrawal, but the effect size only reached the medium level for emotional expression, with women relying more on emotional expression than men (see Table 3). Although our aim was to examine the interaction between gender and personality type, according to the Gendered Innovations, due to the non-significant interaction effect, the main effects need to be reported. Therefore, the results by gender are shown here again, while they were also reported in a previous paper from the same project data collection (currently under review) focused on the effect of gender in stress coping strategies.

Related to personality types, differences were significant for all the dimensions of the CSI, but the effect size did not reach the medium level for problem avoidance (see Figure 1). The post hoc multiple comparisons (see Figure 1) revealed lower problem-solving scores in insecure participants than in some others (sceptic, brooder, entrepreneur, and complicated); higher scores in entrepreneurs than in some others (insecure, brooder, hedonist and impulsive); and also higher scores in complicated participants than in impulsive ones. Self-criticism scores were higher among the insecure than in others (sceptic, hedonist, and entrepreneur); it was also higher in impulsive participants than in others (sceptic and entrepreneur); and lower in entrepreneur participants than in others (insecure, brooder, and complicated). The emotional expression scores of insecure participants were lower than those of impulsive, entrepreneur and complicated ones. The insecure type obtained higher wishful thinking scores than others (sceptic and hedonist); sceptics obtained lower scores than others (insecure, brooder, impulsive and complicated); and the scores of entrepreneur participants were lower than others (insecure, brooder, impulsive, and complicated). Regarding social support, scores were lower among the insecure than among others (impulsive, entrepreneur and complicated) and higher for entrepreneurs than for brooders. Cognitive restructuring scores were lower among the insecure participants than the others (sceptic, hedonist, impulsive, entrepreneurial, and complicated) and higher among entrepreneurs than others (insecure, brooder and impulsive). No post hoc differences were found in problem avoidance scores. Finally, social withdrawal scores were higher among the insecure than among others (hedonist, entrepreneur and complicated) and lower among the entrepreneurs than others (insecure and brooder).

### 3.3. Personality Types and Perceived Coping Efficacy

There was a significant association between personality types and perceived coping efficacy, Welch’s *F*(7, 145.39) = 20.69, *p* < 0.001, with medium effect size, R^2^ = 0.15. The Games–Horwell test (see Table 4) revealed that the insecure type perceived less efficacy in their coping strategies than the sceptic, hedonist, impulsive, entrepreneur and complicated types. Brooder types perceived less efficacy in their coping strategies than hedonists and entrepreneurs. Finally, impulsive types perceived less efficacy in their coping strategies than entrepreneur types.

## 4. Discussion

To the best of our knowledge, this is the second study to examine the association between personality types and (i) coping strategies and (ii) perceived coping efficacy among a large sample of university students using a person-centred approach. The study also explored the distribution of gender in both the personality types and coping strategies.

Along with previous personality types proposed study by Vollrath and Torgersen [12], the most representative types in our sample were the insecure (21.7%), the entrepreneur (17.9%) and the brooder (17.8%), and the less representative types were the spectator (1.6%) and the sceptic (4.8%). However, the distribution of the impulsive, complicated and hedonist types was different between the two studies. Regarding gender, the types most prevalent among women were the insecure (20.8%) and the brooder type (19%), as also found by Vollrath and Torgersen [12]. However, while the most representative types among men in this study were the insecure (25.1%) and the entrepreneur (15.6%), the hedonist and insecure types were most prevalent in the Vollrath and Torgersen [12] sample. In both studies, the most representative types for women were similar, and women were over-represented [12]. It is worth highlighting that the current study focused exclusively on psychology students. In contrast, the earlier study included a cohort of students working toward different degrees. The method for creating the types was different between the two studies. In the Vollrath and Torgersen study [12], each participant was assigned to one of the eight personality types by splitting the scales at the median and combining high and low scores. However, our personality types were created according to the mean T-score obtained in each dimension of the NEO-FFI, following the statistical criteria of scores 0.5 SD above and/or below the normative group mean.

Personality may explain why some people are more vulnerable to stress than others [16] and may influence the reactivity to the stressor, thus affecting the choice of coping strategies, the degree of effectiveness of the chosen coping strategy, or both [47]. The current findings indicate distinct coping strategies among different personality types. Specifically, we found two contrasting profiles in terms of vulnerability to stress and the use of adaptative coping strategies: the insecure and the entrepreneur types.

The insecure type was characterised by a high vulnerability with poor coping since it combines high neuroticism with low conscientiousness and low extraversion. These individuals presented high scores in wishful thinking, social withdrawal, and self-criticism, all of which are considered dysfunctional coping strategies; similarly, they had low scores for problem-solving, cognitive restructuring, emotional expression, and social support (functional coping strategies). In contrast, the entrepreneur was characterised by a low vulnerability to stress since this personality type combines low neuroticism with high conscientiousness and high extraversion. These individuals presented high scores in adaptive coping strategies, such as problem-solving, cognitive restructuring, emotional expression, and social support, and low scores in dysfunctional coping, such as wishful thinking, social withdrawal and self-criticism.

Our results align with the previous study [12], which found that individuals with high neuroticism and conscientiousness—the impulsive and especially insecure types—used maladaptive coping strategies. This could indicate that students with an impulsive and, more significantly, insecure personality type are more vulnerable to stress. In contrast, personality types with low neuroticism and high conscientiousness, particularly the entrepreneur, opted for adaptive coping strategies. Thus, our findings reinforce the importance of personality and coping with student stress, as noted in previous studies [48,49]. Students with an entrepreneurial personality type could choose the right coping strategies and use them effectively to reduce stress experienced in academic situations. Some researchers found that coping strategies allowed students to change the course of things, develop more adaptive behaviours, possibly expand on their academic achievements [50,51] and experience fewer symptoms of depression [52,53]. In contrast, the use of maladaptive coping strategies such as wishful thinking and self-criticism, mainly by the insecure and impulsive types, is a serious problem since there is a negative association between the use of maladaptive coping strategies and academic performance [54] and mental and physical health [55].

As in the study by Vollrath and Torgersen [12], no interaction between gender and personality type was found. That is, men and women with the same personality type chose similar coping strategies and perceived similar coping efficacy. This result is aligned with the universal vision of the FFM [2].

In terms of gender differences in the coping strategies (regardless of the personality types later assigned), our results were like those obtained in other recent investigations, which found significant differences between women and men. Specifically, women were more likely to use emotion-focused engagement strategies like seeking social support and emotional expression, and men were more likely to use avoidance strategies such as problem avoidance and social withdrawal [40,42]. However, this result must be interpreted with caution since among all the coping strategies mentioned, only the relationship between emotional expression and gender obtained a medium effect. Furthermore, it is important to bear in mind that gender differences in coping behaviour are likely due to gender socialisation as opposed to inherent differences in coping behaviours of men and women [41].

Regarding the relationship between personality and perceived coping efficacy, our findings shed more light on this issue. According to our results, personality types characterised by low neuroticism with high conscientiousness—like the entrepreneur and the sceptic—exhibited higher perceived coping efficacy than those with a high score in neuroticism. It is interesting to note that the role of extraversion in perceived coping efficacy did not seem to have much of an effect. The entrepreneur and sceptic personality types were similar regarding the high level of their perceived coping efficacy, with the entrepreneur showing high extraversion and the sceptic low extraversion. Stress coping skills and perceived coping efficacy may be two different concepts, but they are related. Students who cannot resolve a series of problems associated with their university experience can suffer mental stress and frustration associated with academic failure [56]. For instance, if students characterised as insecure are involved in a stressful task, they may not believe they are up to it, leading them to use maladaptive coping strategies [35] and making them vulnerable to chronic stress. Therefore, a better understanding of how individuals with different personality types manage stress, specifically academic stress, could prove invaluable for intervention and prevention efforts designed to enhance academic achievement and bolster psychological well-being and health. Overall, and in accordance with Vollrath and Torgersen [12], our findings suggest that this typology helps address the question as to how individuals with different combinations of personality traits manage stressful situations and their perception of the resources they have to cope with them.

The main strength of this study includes the analysis of the typological personality approach in a large university sample, using a standard and validated measure of personality traits and coping strategies. However, the study has some limitations. First, the results cannot be generalised to the general population or students from other schools and universities since all participants were recruited from the same university, specifically from the School of Psychology. Therefore, future studies are needed with samples more diverse in terms of age, region, and culture. Second, the study design did not enable causal inferences to be made about coping strategies, perceived coping efficacy and personality types, nor did it provide insight into how personality types evolve over time. Therefore, further prospective longitudinal studies are needed. Third, this study relied on a single CSI item to measure perceived coping efficacy. Since this was considered an important variable in controlling stress and is a protective factor against the impact of day-to-day stressors, it should ideally be measured with a specific instrument. Fourth, this study involves at least two generations: those born between 1980 and 1999, considered as Generation Y, and those born after 2000, considered as Generation Z. Some empirical evidence has been found on generational differences in coping strategies [57]. The potential generational effect was not considered in our study.

The main recommendation that emerges from our results is that high levels of neuroticism, especially if associated with low levels of conscientiousness (insecure and impulsive types) or low levels of extraversion (brooder type), are the warning signs of maladaptive academic stress coping and low perceived efficacy. More than half of the psychology students in our sample fit one of these three types.

Future studies should examine other characteristics that may influence the relationship between personality types, coping strategies and perceived coping efficacy in the university context, such as prior academic performance and self-regulation; motivation could also be of interest. Although there is no consensus on the optimal way to determine personality type [6], further studies with more advanced statistical techniques are needed for a definitive scoring process of the eight personality types. The CSI instrument assesses coping strategies used to manage or tolerate stressful situations; the current study has focused only on one type of stressful situation, e.g., academic stress. Students may cope with academic stress differently than they would with other life stressors. Therefore, future studies are necessary to analyse whether the relationship between personality types and coping strategies differ from one stressful situation to the next, as well as longitudinal studies to find out if this relationship remains stable over time or changes as people age. In addition, further research is needed to clarify how gender relates to coping strategies.

## 5. Conclusions

In our sample of mostly female psychology students, we found a high prevalence of insecure, entrepreneur, and brooder types, with no gender differences except for the spectator and hedonistic types, which were more frequent in men. Students with personality types that combine the personality traits of low neuroticism and high conscientiousness (entrepreneur and sceptic types) used the most adaptive coping strategies and presented high levels of perceived coping efficacy. In contrast, the students whose personality types were characterised by high neuroticism and low conscientiousness (insecure and impulsive types) showed maladaptive coping strategies and low levels of perceived coping efficacy. There was no personality-type-gender interaction, nor were there gender differences in perceived coping efficacy; however, there were in the use of coping strategies, with women being more likely to use emotional expression and social support (emotion-focused engagement strategies) and men more likely to use problem avoidance and social withdrawal (avoidance strategies). If the students most at risk of academic stress could be identified, they could be provided with coping strategies to raise their perceived coping efficacy to help them manage academic stress and improve their psychological well-being.

## Figures and Tables

**Figure 1 behavsci-13-00687-f001:**
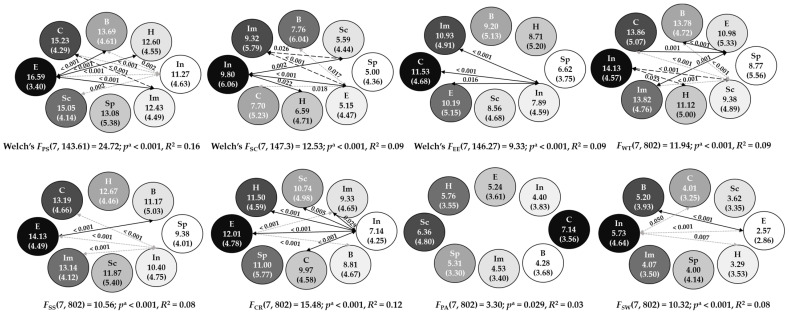
Means (*M*) and Standard Deviations (*SD*) for each Strategy by Personality Type with Adjusted Probabilities (*p*^a^) for Post hoc Comparison Tests. Arrows represent significant (*p* < 0.05) post hoc comparisons. Personality types appear from higher means (darker colour) to lower means (lighter colour). Sp = Spectator, In = Insecure, Sc = Sceptic, B = Brooder, H = Hedonist, Im = Impulsive, E = Entrepreneur, C = Complicated, PS = Problem Solving, SC = Self-Criticism, EE = Emotional Expression, WT = Wishful Thinking, SS = Social Support, CR = Cognitive Restructuring, PA = Problem Avoidance, SW = Social Withdrawal.

**Table 1 behavsci-13-00687-t001:** The Eight Personality Types [11,12].

Type Label	Neuroticism	Conscientiousness	Extraversion
Spectator	Low	Low	Low
Sceptic	Low	High	Low
Hedonist	Low	Low	High
Entrepreneur	Low	High	High
Insecure	High	Low	Low
Brooder	High	High	Low
Impulsive	High	Low	High
Complicated	High	High	High

**Table 2 behavsci-13-00687-t002:** Descriptive Statistics for Personality Types for Total Sample and by Gender.

		Sp	In	Sc	B	H	Im	E	C
Men	N (%)	9 (5.4)	42 (25.1)	10 (6)	21 (12.6)	25 (15)	22 (13.2)	26 (15.6)	12 (7.2)
Women	N (%)	4 (0.6)	134 (20.8)	29 (4.5)	123 (19.1)	33 (5.1)	98 (15.2)	119 (18.5)	103 (16.0)
Total	N (%)	13 (1.6)	176 (21.7)	39 (4.8)	144 (17.8)	58 (7.2)	120 (14.8)	145 (17.9)	115 (14.2)

Note: Sp = Spectator, In = Insecure, Sc = Sceptic, B = Brooder, H = Hedonist, Im = Impulsive, E = Entrepreneur, C = Complicated.

**Table 3 behavsci-13-00687-t003:** ANOVA Tests Results with Adjusted Probabilities for each Coping Strategy by Gender.

Coping	Men *M* (SD)	Women *M* (SD)	*F*/Welch’s *F*	df	*p*	*p* ^a^	*R* ^2^
PS	13.47 (4.97)	13.75 (4.65)	0.48	1808	0.488	>0.999	<0.01
SC	8.00 (5.85)	7.65 (5.66)	0.49	1808	0.486	>0.999	<0.01
EE	7.14 (4.37)	10.21 (5.02)	61.31 **	1290.66	<0.001	<0.001	0.06
WT	12.26 (5.27)	13.05 (5.08)	3.21	1808	0.074	>0.999	<0.01
SS	10.81 (5.14)	12.59 (4.70)	18.20 **	1808	<0.001	<0.001	0.02
CR	9.98 (5.14)	9.48 (4.83)	1.40	1808	0.237	>0.999	<0.01
PA	5.56 (3.73)	4.48 (3.70)	11.35 *	1808	0.001	0.013	0.01
SW	5.12 (4.23)	4.06 (3.75)	10.06 *	1808	0.002	0.025	0.01

Note: Sp = Spectator, In = Insecure, Sc = Sceptic, B = Brooder, H = Hedonist, Im = Impulsive, E = Entrepreneur, C = Complicated; PS = Problem Solving, SC = Self-Criticism, EE = Emotional Expression, WT = Wishful Thinking, SS = Social Support, CR = Cognitive Restructuring, PA = Problem Avoidance, SW = Social Withdrawal; ^a^ = adjusted *p* (*p* × 16); * adjusted *p* < 0.05, ** adjusted *p* < 0.01.

**Table 4 behavsci-13-00687-t004:** Descriptive Statistics of Perceived Coping Efficacy and Probability in the Games–Horwell Test for each Pairwise Comparison.

	M	SD	Sp	In	Sc	B	H	Im	E
Sp	2.77	1.09							
In	1.98	1.07	0.261						
Sc	3.10	0.94	0.971	<0.001 **					
B	2.33	1.22	0.859	0.116	0.002 **				
H	2.91	1.01	1.000	<0.001 **	0.981	0.016 *			
Im	2.39	1.15	0.927	0.042 *	0.005 **	1.000	0.051		
E	3.25	1.02	0.784	<0.001 **	0.990	<0.001 **	0.408	<0.001 **	
C	2.77	1.04	1.000	<0.001 **	0.597	0.039 *	0.990	0.136	0.006 **

Note: Sp = Spectator, In = Insecure, Sc = Sceptic, B = Brooder, H = Hedonist, Im = Impulsive, E = Entrepreneur, C = Complicated; * *p* < 0.05, ** *p* < 0.01.

## Data Availability

The data that support the findings of this study are available from the corresponding author, F.J.C.-G., upon reasonable request.

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
