# Peer review of "Effective Coping with Academic Stress Is a Matter of Personality Types: Revisiting the Person-Centred Approach"

_behavsci, 2023, doi:10.3390/bs13080687_

Round 1
Reviewer 1 Report
Previous note
The present article has the potential to be published, after some changes in the original text and updating of the bibliography. The topic addressed is of great interest to research in general and, more specifically, in the area of Educational Psychology, especially those who are dedicated to studying how the different dimensions of personality can influence or be associated with other constructs in the fight against academic stress. Entering the university academic world is seen by some students as an opportunity to manage their activities, explore new environments, and build new relationships. However, for others, it is perceived as potentially anxiogenic and a promoter of stress-inducing situations. The concept of coping has been described as the set of strategies the individual uses to adapt to adverse or stressful events. In recent years, research has turned to convergences between coping and personality. This trend has been motivated, in part, by the cumulative body of evidence that indicates that situational factors are not able to explain all the variations in the coping strategies used by individuals. On the other hand, the interest aroused by the scientific credibility of studies on personality traits, in particular the Big Five model, has expanded studies in this direction.
Thus, the reflections developed in this article can trigger new reflective moments and investigations on the importance of the role of the constructs now under study to overcome the underlying risks of stress in a university environment, by opting for the use of more adaptive coping strategies, helping them to manage academic stress and improve your psychological well-being.
In this sequence, Behavioral Sciences (BS), through the publication of this article, has the possibility of contributing to the enrichment of the theoretical foundation of the subject under study and to the promotion of the creation of intervention programs that can help to develop strategies for emotionally overcoming regulated in order to deal with the different situations experienced in everyday life.
In addition, it is essential that the results obtained in this study are duly clarified and substantiated, with a view to valuing the analyzed article. It should also be noted that the text under analysis is intended for a diverse academic population with different knowledge on the subject under study, and should be structured in such a way as to enrich the readers' understanding.
Thus, it is essential that the production of BS, as a scientific journal, maintains its quality levels in the area under study. In this sense, some suggestions for changes to the article presented are indicated, namely:
Abstract:
1. Introduction
- The statement “A limitation of the existing literature on personality and stress and coping strategies is its analysis of individual personality traits in isolation, without any considerations of how these operate together” by itself seems a bit ambiguous, it is based on a publication by 2008, the last reference of the literature review of the present study is from 2020, and bibliographical references in the last 5 years are practically non-existent. It is suggested to reformulate the statement and present more recent studies on the topic under study, including those from 2023. I recall that, in the context of personality studies, the Big Five should be one of the most used instruments in research worldwide.
- It is suggested to further develop the concepts of personality traits and coping styles and strategies, in relation to their different theoretical and methodological positions, in order to better understand the effectiveness and possible relationships between these two constructs. This development will help to interpret and compare the final results obtained in this study.
- The “Introduction” needs further literature review. Many of the referred statements refer to studies with more than two decades.
- Present some empirical evidence that addresses the association between personality types (spectator, sceptic, hedonist, entrepreneur, insecure, brooder, impulsive, complicated) and the subscales of the CSI dimensions, in order to compare the final results of the study. The same evidence regarding the association with gender.
2. Materials and Methods
2.1. Participants
- Justify the reason for the difference in the initial sample of 1305 students to be reduced to 810.
- Due to the fact that the sample presents a large difference between women and men, it would be interesting to know the average age of each gender, perhaps for possible conclusions in the presentation of the results.
- Justify the reason why only Psychology students were selected, with the opportunity for the sample to be broader to students from other courses.
2.2. Measures
- Considering current standards and research efforts to build smaller scales for a variety of reasons, can NEO-FFI be considered a small inventory?
- Part of the operationalization of the instruments could be presented in the “Introduction” part when developing the constructs under study, as previously mentioned. Here, only the constitution of the instruments could be described (items, dimensions, psychometric values,…).
2.3. Procedure
2.4. Data Analyses
3. Results
3.1. Descriptive analyses
4. Discussion
- As previously mentioned, eventually a more recent literature review will find other results that can be compared with those obtained in this study.
5. Conclusions
References
- Update the bibliography with more current references.
Reviewer 2 Report
The present research work carried out by experts in the field (Dept of Personality and Dept Experimental Psychology) addresses the role of (person-centered) personality traits on coping strategies in academic stress and their perceived efficacy in a large sample of students in a Spanish university. This is a relevant topic that urges scientific and academic efforts as it is dramatically increasing in recent years, as reported by different academic and research reports from different countries. The role of gender is also evaluated, but it probably yields fewer effects than expected.
In the following paragraphs, this reviewer would like to indicate, in a constructive manner, some aspects that need some improvement, clarification or that could benefit the final version.
1) The work's strength is flattened in the title and abstract, and these sections could benefit from straightforward writing. For instance, the title does not follow the conceptual structure of the research performed (refers to personality, coping, and then gender), whereas the hypothesis considers person-centred approach for personality traits and gender as two key factors, predictors explaining coping strategies in academic stress and perceived efficacy. On the other hand, the title should not present the question of interest (study the relationship between X factors in a Y output), but be a home take message of the findings. Please, reconsider.
2) The real academic scenario is worrisome; therefore, I strongly suggest that the first introductory sentences of the abstract that refer to the reasons for the present work be targeting this increasing (if not alarming) situation. Please, reconsider.
·
3) The aims referred to two important questions. The first, coping strategies, more common in this field of research, and the second, quite interesting, the perceived coping efficacy. The authors should emphasize this second one, too (title, etc)
4) From the aims, it was expected that factor interactions (PT x G) would appear. Discussion about the gender factor is given, but not the factor interaction. Please, discuss.
·
5) The abstract provides extensive details on the statistical tools used, but nothing is said about an interesting aspect of this work, as perfectly referred to in the introduction, about the person-centered as compared to the classical variable-centered approach. This should also be highlighted in the title, as the approach is part of the strong points of the present work.
6) Table 1 will benefit from presenting the data as N (%) instead of using two rows, similar to what is done in Table 3. Since tables or figures should be interpretable per se, sample size of each gender should be added to the table or legend.
7) Please, use colors instead of greys in figure 1. Also please, note, that the ‘strategy’ can hardly be read, so I suggest adding a heading in each one. Then, the name can be omitted as a subindex after the F and that would also help to make the statistical text more compacted in only one raw.
·
8) Significant gender differences were found in Personality types and Coping strategies, but after several analysis the main conclusion that can be given was quite reductionist i.e. ‘only reached medium level for emotional expression’.
9) Limitations are well detected and discussed. Still, nothing is said with regards to the generational effect (centennials, millennials, etc)
·
10) I strongly suggest to do conclusive remarks that emphasize all the findings, as now they are focused on the two typologies (combination of high/low traits) but do not mention the different distribution found in personality types (more and less frequent).
11) A part of the obvious (but still needed to be mentioned) fact that students are more at risk of academic stress should be identified, the work will benefit from future directions and/or recommendations based on the present findings.
12) From the last sections of the Ms, it can be noted that the work was done without receiving external finding, which means the scientific and academic commitment of the authors is commendable.
Round 2
Reviewer 2 Report
The authors have provided a detailed point-by-point response to all queries and suggestions made by this reviewer, and have done a number of modifications in several parts of the Ms to adapt it to those aspects. The most important one refers to the abstract, that has been rewritten to give the most of their work. The new statements in the introduction emphasize the relevance of the scenario among young people, mostly students and will help to raise also concern and value to the present work. Gender issues have been improved and conclusions are now more clear in this respect.
The title is more goal directed. Just, please, check the title, since "types" in "Effective coping with academic stress is a matter of types:" should refer to "personality types" and then the title could read "Effective coping with academic stress is a matter of personality types: revis-2 iting the person-centred approach."
Author Response
The authors have provided a detailed point-by-point response to all queries and suggestions made by this reviewer, and have done a number of modifications in several parts of the Ms to adapt it to those aspects. The most important one refers to the abstract, that has been rewritten to give the most of their work. The new statements in the introduction emphasize the relevance of the scenario among young people, mostly students and will help to raise also concern and value to the present work. Gender issues have been improved and conclusions are now more clear in this respect.
We are grateful for the comments by Reviewer 2, which have contributed to highly improve the manuscript.
The title is more goal directed. Just, please, check the title, since "types" in "Effective coping with academic stress is a matter of types:" should refer to "personality types" and then the title could read "Effective coping with academic stress is a matter of personality types: revisiting the person-centred approach."
This is an excellent proposal because increases the title specificity. Thank you very much.